# Inhibition of USP2 Enhances TRAIL-Mediated Cancer Cell Death through Downregulation of Survivin

**DOI:** 10.3390/ijms241612816

**Published:** 2023-08-15

**Authors:** Tak Gyeom Lee, Seon Min Woo, Seung Un Seo, Shin Kim, Jong-Wook Park, Young-Chae Chang, Taeg Kyu Kwon

**Affiliations:** 1Department of Immunology, School of Medicine, Keimyung University, Daegu 42601, Republic of Korea; tg1915@naver.com (T.G.L.); woosm724@gmail.com (S.M.W.); sbr2010@hanmail.net (S.U.S.); god98005@dsmc.or.kr (S.K.); j303nih@dsmc.or.kr (J.-W.P.); 2Research Institute of Biomedical Engineering and Department of Cell Biology, School of Medicine, Catholic University of Daegu, Daegu 42472, Republic of Korea; ycchang@cu.ac.kr; 3Center for Forensic Pharmaceutical Science, Keimyung University, Daegu 42601, Republic of Korea

**Keywords:** ML364, USP2, TRAIL, deubiquitinase, survivin

## Abstract

Ubiquitin-specific protease 2 (USP2) is a deubiquitinase belonging to the USPs subfamily. USP2 has been known to display various biological effects including tumorigenesis and inflammation. Therefore, we aimed to examine the sensitization effect of USP2 in TRAIL-mediated apoptosis. The pharmacological inhibitor (ML364) and siRNA targeting *USP2* enhanced TNF-related apoptosis-inducing ligand (TRAIL)-induced cancer cell death, but not normal cells. Mechanistically, USP2 interacted with survivin, and ML364 degraded survivin protein expression by increasing the ubiquitination of survivin. Overexpression of survivin or USP2 significantly prevented apoptosis through cotreatment with ML364 and TRAIL, whereas a knockdown of *USP2* increased sensitivity to TRAIL. Taken together, our data suggested that ML364 ubiquitylates and degrades survivin, thereby increasing the reactivity to TRAIL-mediated apoptosis in cancer cells.

## 1. Introduction

The ubiquitin proteasome system (UPS) is a major pathway of protein quality control and plays an important role in diverse cellular actions including proliferation, DNA replication, and apoptosis. UPS is a process wherein target protein is degraded through 26S proteasomal activation by attaching mono- or poly-ubiquitin [1]. The complex containing E1-activating, E2-conjugating, and E3-ligase enzymes catalyzes ubiquitination, whereas deubiquitinases (DUBs) prevents ubiquitination [2,3,4]. Approximately 100 DUBs have been characterized and are categorized into two classes, including cysteine proteases and metalloproteases [5]. Several small molecule inhibitors of the DUBs could be appropriately developed and designed into an efficacious anticancer drug. 

Ubiquitin-specific protease 2 (USP2) is a cysteine protease which is a well-characterized member of the USP family [6]. It acts as a crucial regulator for physiological and pathological processes such as carcinogenesis, inflammation, and circadian rhythm regulation [7]. A high level of USP2 is observed in prostate, hepatoma, bladder, and glioma cancers [8,9,10,11,12]. ML364, a small molecule inhibitor of USP2, causes cell cycle arrest through degradation of cyclin D1 in colorectal cancer [13]. These effects of ML364 support the possibility of USP2 as a therapeutic target in cancer therapy. Moreover, USP2 modulates the susceptibility to antineoplastic agents in prostate cells [14]. Even though some researchers have reported that ML364 exerts the broad-spectrum anti-virulence effect [15], the underlying specific molecular mechanisms of ML364 on cancer cell death remain unclear.

A tumor necrosis factor (TNF)-related apoptosis-inducing ligand (TRAIL) triggers apoptosis of cancer cells through DISC formation by binding to death receptors (DRs) and recruiting caspase-8 and FADD. Caspase-3 activation is directly required for apoptosis in type I cells, and the truncated Bid-dependent mitochondrial pathway is involved in apoptosis of type II cells [16]. Despite it having the selective capability of causing death in tumors, several tumors indicate a resistance to TRAIL because of DRs downregulation or Bcl-2 and IAP proteins’ upregulation [17]. Therefore, studies to overcome the issue of TRAIL resistance are required for cancer therapy.

In this study, we identified the new target substrate of USP2 and examined the effect of ML364 as a TRAIL sensitizer.

## 2. Results

### 2.1. The USP2 Inhibitor ML364 Induces Downregulation of Survivin Expression

First, we examined the change in apoptosis-related proteins by a USP2-specific inhibitor, ML364, in Caki-1 cells. Only survivin was decreased in ML364 treatment, whereas other proteins were not changed (Figure 1A). Also, when we examined the effect of ML364 on diverse cancer cell lines (renal carcinoma, ACHN; prostate carcinoma, DU145; and lung carcinoma, A549), all tested cells showed survivin downregulation in ML364 treatment (Figure 1B). To exclude the off-target effects of ML364, we depleted *USP2* expression using specific siRNA. The knockdown of *USP2*-induced survivin downregulation is presented in Figure 1C. To certain that the USP2 contributes to survivin protein regulation, we performed an overexpression system of USP2 wild-type (WT) in *USP2*-knockdowned cells. We observed that USP2 WT rescues survivin downregulation through *USP2* siRNA (Figure 1D). Therefore, these data showed that pharmacological and genetic inhibition of USP2 downregulate survivin expression in many cancer cells. 

### 2.2. USP2 Regulates Survivin Stability

Even though ML364 inhibited the survivin protein level, the expression of survivin mRNA was not altered by ML364 (Figure 2A,B). Next, we explored the stabilization of survivin protein by ML364 using cycloheximide (CHX), an inhibitor of protein synthesis. ML364 or *USP2* siRNA in the presence of CHX further decreased survivin protein levels compared to CHX alone (Figure 2C,E). In addition, a proteasome inhibitor MG132 prevented ML364-mediated survivin degradation (Figure 2D). To verify the involvement of catalytic activity of USP2 in survivin stabilization, we transfected a catalytic mutant (C276A) of USP2 plasmid in Caki-1 cells. Overexpression of USP2 WT sustained survivin expression, whereas overexpression of USP2 C276A further degraded survivin expression compared to vector-transfected cells (Figure 2F). Therefore, these results indicated that survivin protein stability is modulated by the expression level of USP2.

### 2.3. USP2 Interacts with and Deubiquitinates Survivin

To investigate the interaction between USP2 and survivin, we performed immunoprecipitation (IP) assay using a USP2 antibody. USP2 bound to survivin (Figure 3A). Further, we examined survivin ubiquitination and found an increase in polyubiquitination of survivin in ML364 treatment (Figure 3B). These data showed that inhibition of USP2 destabilizes survivin through the increase in ubiquitination.

### 2.4. ML364 Sensitizes TRAIL-Mediated Apoptosis

Since ML364 induces survivin downregulation, we confirmed the sensitization effect of ML3264 in TRAIL-mediated cell death. A sub-lethal dosage of ML364 induced cleavage of poly (ADP-ribose), polymerase (PARP), and sub-G1 populations in various TRAIL-treated cancer cells (Figure 4A). However, combinations of ML364 and TRAIL did not indicate morphological change and accumulation of sub-G1 populations in normal human mesangial cells or human endothelial cells (EA.hy926) (Figure 4B). These data revealed that ML364 effectively sensitizes cancer cells to TRAIL-mediated apoptosis.

### 2.5. Downregulation of Survivin Contributes to ML364-Mediated TRAIL Sensitization

Combined treatment with ML364 and TRAIL induced nuclear condensation and DNA fragmentation (Figure 5A,B). As a result of examining the relevance of caspase activation in the combined treatment, an increase in caspase-3 activity (DEVDase) was examined, but no increase in activity was seen in the individual treatment (Figure 5C). A pan-caspase inhibitor, z-VAD-fmk, completely blocked the combined-treatment-induced apoptosis (Figure 5D). Since we identified that ML364 decreases survivin expression (Figure 1A), we progressed the overexpression system to verify the involvement of survivin in ML364-plus-TRAIL-induced apoptosis. Overexpression of survivin inhibited apoptosis by combined treatment (Figure 5E). These data suggested that survivin contributes to depletion of USP2-mediated TRAIL sensitization.

### 2.6. USP2 Plays an Important Role in TRAIL-Mediated Apoptosis 

We explored the sensitivity to TRAIL using *USP2* siRNA instead of ML364. As shown in Figure 6A, when *USP2* was depleted by siRNA, TRAIL accumulated a sub-G1 population and cleaved PARP in various cancer cells. In addition, we observed that the ectopic of USP2 WT diminishes the apoptosis, whereas overexpression of catalytically mutated USP2 (C276A) causes apoptosis in ML362-plus-TRAIL-treated cells (Figure 6B). These results demonstrated that inhibition of USP2 plays an important role in the increase in TRAIL sensitization.

## 3. Discussion

We suggested survivin as a novel substrate of USP2 and showed that ML364-mediated decrease in USP2 activity is involved in TRAIL sensitization through degradation of survivin. These data demonstrated that ML364 can be an attractive TRAIL sensitizer.

Renal cell carcinoma (RCC) is the 10th most common cancer worldwide, and is classified according to pathologic and molecular characteristics including clear cell, papillary, or chromophobe RCC. Above all, ccRCC (e.g., Caki-1 cells) is accounted for in kidney-cancer-related deaths and is mutated in the von Hippel–Lindau (~90%), PBRM1 (~50%), and BAP1 (~15%) genes [18]. Primary RCC can be completely recovered from through surgery, whereas metastatic RCC is difficult. Therefore, targeted drugs against VEGF and mTOR signaling pathways are developed for treatment of metastatic RCC [19]. It is refractory to targeted or chemotherapeutic agents because of toxicity and resistance [20]. Thus, to overcome resistance depending on gene mutation and chemotherapy can higher the death of RCC. Our data suggested that pharmacological inhibition (ML364) and depletion of *USP2* sensitizes TRAIL-mediated apoptosis in metastatic ccRCC (Caki-1 cells) and metastatic pRCC (ACHN cells) (Figure 4A and Figure 6A). Taken together, our results provided the potential of USP2 as a therapeutic target for treatment of RCC.

USP2 has been associated with tumorigenesis and high reported expression levels in malignant tumors [8,9,10,11,12]. Multiple studies have reported several molecular targets of USP2 in cancer cells. For example, USP2 interacts and stabilizes fatty acid synthase [8], MDM2 [21], MDMX [22], Cyclin D1 [23], Aurora-A [24], Smad4 [25], and β-catenin [26]. USP2 knockdown via siRNA inhibits actinomycin and TNF-α-induced apoptosis via c-FLIP upregulation in hepatocytes [27]. *USP2* siRNA decreases Itch expression, a ubiquitin-ligase which is a negative regulator of c-FLIP. In our study, c-FLIP did not upregulate through the USP2 inhibitor ML364 (Figure 1A). This contradiction may be attributed to differences between the cell lines that were used. However, the involvement of other c-FLIP regulatory pathways in ML364-treated cells cannot be completely ruled out. Furthermore, USP2 is resistant to drugs in both immortalized and transformed prostate cells. Overexpression of USP2 defends from cisplatin-mediated oxidative stress by producing glutathione [14]. Although USP2 is a regulator of oncogenic behavior in cancer through regulating protein stability of diverse substrates, USP2-targeted substrates and the underlying mechanisms of cancer cell death remain unclear. 

To identify novel substrates of USP2 on apoptosis in cancer, we surveyed alteration of apoptotic regulatory proteins stemming from ML364 treatment. We found that ML364 only induces survivin downregulation (Figure 1A). In addition, USP2 interacted with survivin and inhibition of USP2 by ML364-induced ubiquitination of survivin (Figure 3A,B).

Survivin is regulated at the transcriptional and post-translational levels. Ubiquitination and degradation of survivin is induced by many E3 ubiquitin ligases, such as FBXL7, CUL9, and XIAP [28,29,30]. DUBs also participate in survivin stabilization. For example, extensive evidence has indicated that survivin is deubiquitinated by several DUBs, such as USP1, USP35, USP9X, and STAMBPL1 [31,32,33,34]. We newly identified the USP2 as DUB of survivin. Moreover, we proved using the catalytic mutant USP2 C276A, that USP2 can stabilize survivin through catalytic activation (Figure 2F). In addition, overexpression of USP2 C276A still provoked apoptosis by combination treatment of ML364 and TRAIL, but not USP2 WT (Figure 6B). Therefore, we identified survivin as a novel substrate of USP2, and found that USP2-depletion-mediated survivin degradation contributed to TRAIL sensitization of cancer cells. 

Collectively, our findings indicated that depletion of USP2 decreases survivin ubiquitination, thereby inducing TRAIL-mediated apoptosis.

## 4. Materials and Methods

### 4.1. Cells and Chemicals

Caki-1, ACHN, DU145, and A549 cells were cultured in Dulbecco’s modified Eagle’s medium (Welgene, Gyeongsan, South Korea) containing 10% fetal bovine serum (FBS) (Welgene), 5% penicillin/streptomycin (Welgene), and 100 μg/mL gentamicin (Thermo Fisher Scientific, Waltham, MA, USA) and purchased from the American Type Culture Collection (Manassas, VA, USA). ML364 was obtained from Cayman Chemical Co. (Ann Arbor, MI, USA). Cycloheximide and MG132 were purchased from Sigma-Aldrich (St. Louis, MO, USA) and Calbiochem (San Diego, CA, USA), respectively. Human recombinant TRAIL and z-VAD-fmk were obtained from R&D Systems (Minneapolis, MN, USA).

### 4.2. Transfection

The cells were transfected with the green fluorescent protein (control) and USP2 siRNA using Lipofectamine RNAiMAX (Thermo Fisher Scientific). The siRNAs were obtained from Bioneer (Daejeon, South Korea).

### 4.3. Examination of Protein and mRNA Level

Western blotting was performed on various cancer cell lines to investigate the alteration of protein expression as described previously [35,36]. The harvested cells were lysed using RIPA lysis buffer and separated by sodium dodecyl sulfate–polyacrylamide gel electrophoresis. The protein was transferred onto nitrocellulose membrane (GE Healthcare Life Science, Pittsburgh, PO, USA) and were incubated with primary antibodies overnight, and then the secondary antibody was incubated at room temperature for 2 h. Finally, expression of proteins was detected by an enhanced chemiluminescence kit (Merck Millipore, Darmstadt, Germany). The information on primary antibodies was provided as below: anti-Bcl-2 and anti-DR4 from Abcam (Waltham, MA, USA); anti-Mcl-1 and anti-cIAP2 from Santa Cruz Biotechnology (Santa Cruz, CA, USA); anti-Bax, anti-Bim, and anti-XIAP from Biosciences (San Jose, CA, USA); anti-survivin from R&D System; anti-Bcl-xL, anti-cIAP1, anti-DR5, anti-PARP, anti-USP2, and anti-cleaved caspase-3 from Cell Signaling Technology (Beverly, MA, USA); anti-c-FLIP and anti-caspase3 from Enzo Life Sciences (San Diego, CA, USA). RT-PCR and quantitative PCR were used to analyze mRNA expression, and primer sequences were described previously [37].

### 4.4. Immunoprecipitation and Ubiquitination Assays

Experimental methods were described in a previous study [38]. In brief, the sonicated cell lysates in RIPA lysis buffer containing 10 mM N-ethylmaleimide (Sigma-Aldrich) and 1 mM phenylmethylsulfonyl fluoride (Sigma-Aldrich) were incubated overnight with anti-survivin antibody (Cell Signaling Technology) and subsequently with protein PLUS-Agarose (Santa Cruz Biotechnology) for 2 h at 4 °C. Immunoprecipitation assay was performed with Western blotting using specific primary antibodies, and ubiquitination assay was detected using horseradish peroxidase-conjugated anti-Ub (Enzo Life Sciences) under denaturation conditions.

### 4.5. Certification of Apoptosis

To investigate the apoptosis, we utilized various experimental methods. For sub-G1 analysis, we fixed the cells using 100% ethanol over 1 h. Cells were then stained with propidium iodide (Sigma-Aldrich) after incubation with RNase for 30 min at 37 °C, and then measured by flow cytometry (BD Biosciences, San Jose, CA, USA). For measurement of caspase-3 (DEVDase) activation, cell lysates were incubated with an acetyl-Asp-Glu-Val-Asp p-nitroanilide (Ac-DEVD-pNA), and their activity was measured by a spectrophotometer. To detect nuclear damage, we conducted the 4′, 6′-diamidino-2-phenylindole assay (Roche, Mannheim, Germany) and used the Cell Death Detection ELISA Plus kit (Roche, Basel, Switzerland). Fluorescence images were analyzed using fluorescence microscopy (Carl Zeiss, Jena, Germany). 

### 4.6. Statistical Analysis

The data were analyzed using one-way ANOVA and post hoc comparisons (Student-Newman-Keuls) using the Statistical Package for Social Sciences software (version 22.0; SPSS Inc., Chicago, IL, USA). 

## 5. Conclusions

USP2, a deubiquitinase, plays an important role in many biological processes including cell proliferation, tumorigenesis, and inflammation. Here, we explored the effect of TRAIL sensitization and molecular mechanisms of ML364 in cancer cells. The pharmacological inhibitor (ML364) and siRNA of USP2 downregulated survivin expression at the post-translational level. In addition, USP2 bound to survivin, and ML364 increased ubiquitination of survivin. Furthermore, ML364 treatment and knockdown of USP2 enhanced TRAIL-mediated apoptosis in cancer cells, but not in normal cells. Overexpression of survivin prevented combined treatment with ML364 plus TRAIL-induced apoptosis. These data demonstrated that inhibition of USP2 sensitizes TRAIL-mediated apoptosis through degradation of survivin.

## Figures and Tables

**Figure 1 ijms-24-12816-f001:**
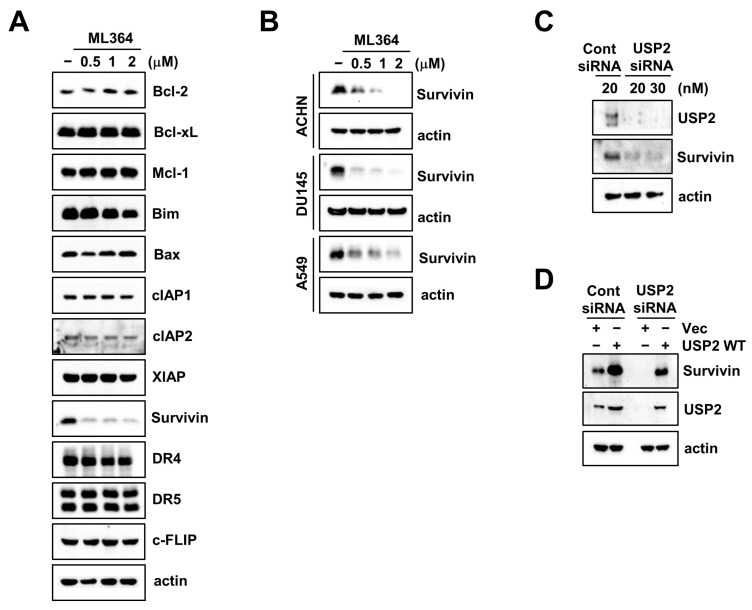
Knockdown of *USP2* induces survivin downregulation in cancer cells. (**A**,**B**) Caki-1 (**A**), ACHN, DU145, and A549 (**B**) cells were treated with various concentrations of ML364. (**C**) Caki-1 cells were transfected with control or USP2 siRNA for 24 h. (**D**) Caki-1 cells were transfected with Cont siRNA or *USP2* siRNA without/with USP2 plasmid for 24 h. The protein expression was detected using Western blotting.

**Figure 2 ijms-24-12816-f002:**
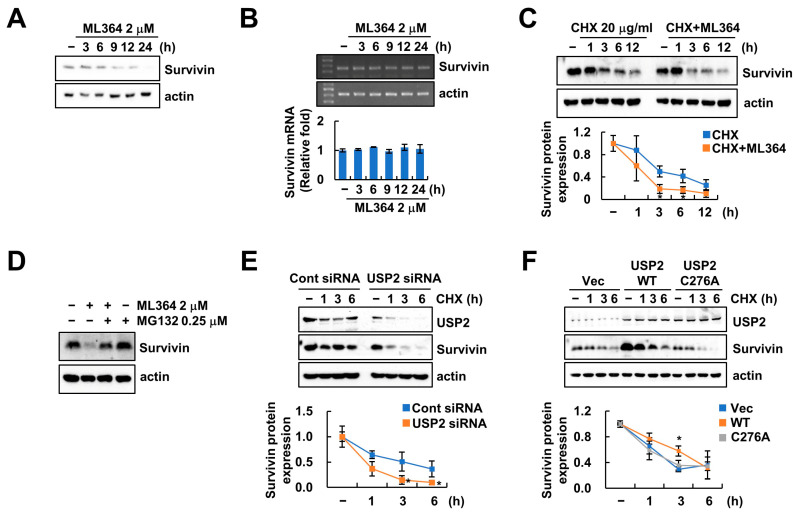
ML364 induces proteasome-mediated survivin degradation. (**A**,**B**) Caki-1 cells were treated with 2 μM ML364 for the different times. (**C**) Caki-1 cells were treated with 2 μM ML364 after pretreatment with 20 μg/mL cycloheximide (CHX) for the different times. (**D**) Caki-1 cells were treated with 2 μM ML364 after pretreatment with 0.25 μM MG132 for 24 h. (**E**) Caki-1 cells were transfected with control or USP2 siRNA and then treated with 20 μg/mL CHX for the different times. (**E**,**F**) Caki-1 cells were transfected with vector, USP2 WT, or mutant (C276A) plasmid for 24 h and then treated with 20 μg/mL CHX for the different times. The protein expression was detected using Western blotting (**A**,**C**–**F**). The c-FLIP mRNA level was detected using RT-PCR and qPCR (**B**). The values in the graph represent the mean ± SD of three independent samples. * *p* < 0.05 compared to the CHX.

**Figure 3 ijms-24-12816-f003:**
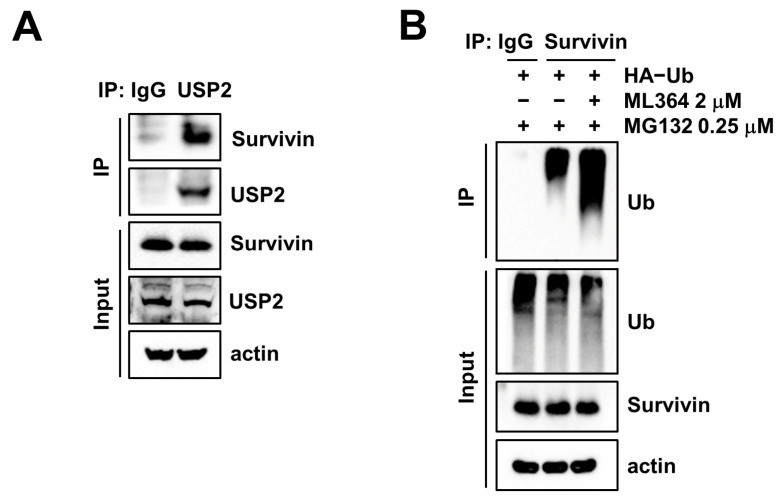
USP2 binds and deubiquitinates survivin. (**A**) Examination of the interaction between proteins using immunoprecipitation (IP). (**B**) Caki-1 cells were transfected with HA-Ub plasmid and treated with 2 μM ML364 after pretreatment with 0.25 μM MG132 for 24 h. These assays were detected using Western blotting.

**Figure 4 ijms-24-12816-f004:**
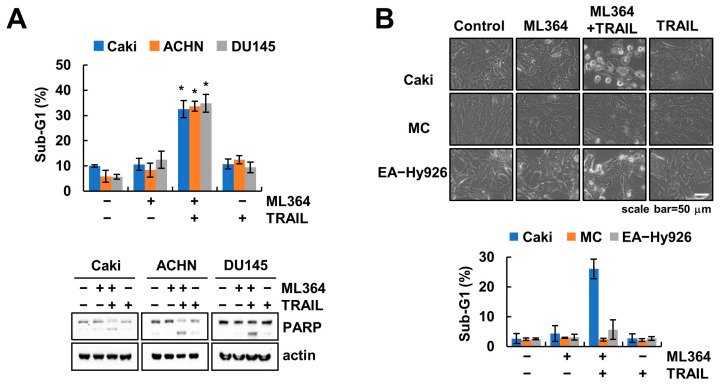
ML364 sensitizes TRAIL-mediated apoptosis in cancer cells. (**A**,**B**) Cancer (**A**,**B**) and normal (**B**) cells were treated with 2 μM ML364 and/or 50 ng/mL TRAIL for 24 h. The sub-G1 population (**A**) and protein expression (**A**,**B**) were determined using flow cytometry and Western blotting, respectively. Cell morphology was assessed using a microscope. Scale bar: 50 μm (**B**). Values in the graphs (**A**,**B**) represent the mean ± SD of three independent experiments. * *p* < 0.01 compared to the control.

**Figure 5 ijms-24-12816-f005:**
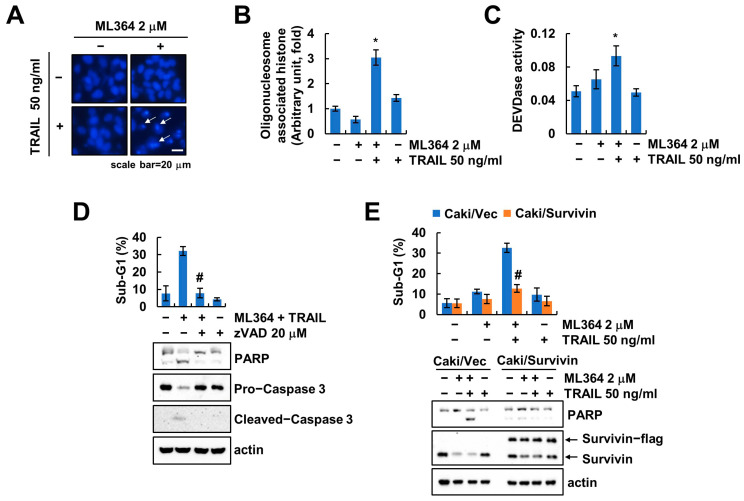
ML364-induced survivin downregulation contributes to TRAIL sensitization. (**A**–**C**) Caki-1 cells were treated with 2 μM ML364 and/or 50 ng/mL TRAIL for 24 h. DNA fragmentation was analyzed using DAPI staining (**A**) and kit (**B**). Measurement of DEVDase (caspase-3) activity using substrate (**C**). (**D**) Caki-1 cells were treated with combination of 2 μM ML364 and 50 ng/mL TRAIL after pretreatment with 20 μM zVAD for 24 h. (**E**) Vector and survivin-overexpressed Caki-1 cells were treated with 2 μM ML364 and/or 50 ng/mL TRAIL for 24 h. The sub-G1 population and protein expression were determined using flow cytometry and Western blotting, respectively (**D**,**E**). Values in the graphs (**B**–**E**) represent the mean ± SD of three independent experiments. * *p* < 0.01 compared to the control. # *p* < 0.01 compared to combinations of ML364 and TRAIL. scale bar: 20 μm. White arrows indicate nucleus condensation.

**Figure 6 ijms-24-12816-f006:**
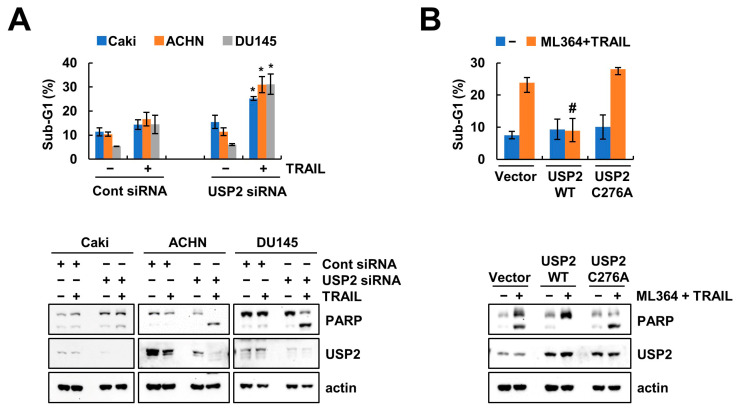
Knockdown of USP2 sensitizes TRAIL-mediated apoptosis. (**A**) Cancer cells were treated with 50 ng/mL TRAIL after transfection of control or USP2 siRNA for 24 h. (**B**) Caki-1 cells were treated with combinations of 2 μM ML364 and 50 ng/mL TRAIL after transfection of vector and USP2 plasmids (WT and C276A) for 24 h. The sub-G1 population and protein expression were determined using flow cytometry and Western blotting, respectively (**A**,**B**). Values in the graphs (**A**,**B**) represent the mean ± SD of three independent experiments. * *p* < 0.01 compared TRAIL treatment in control siRNA. # *p* < 0.01 compared to ML364 and TRAIL treatment in vector.

## Data Availability

The data presented in this study are available on request from the corresponding author.

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
