# Peer review of "Inhibition of USP2 Enhances TRAIL-Mediated Cancer Cell Death through Downregulation of Survivin"

_ijms, 2023, doi:10.3390/ijms241612816_

Round 1

Reviewer 1 Report

This study identified surviving as a new target substrate of USP2 and examined USP2 inhibitor ML364 as a sensitizer in TRAIL-mediated apoptosis. Overall, the finding presented here is interesting, but a big improvement of the manuscript is still required before its acceptance.

Below are some specific comments.

1.       The introduction part is short, it lacks the brief introduction of TRAIL-mediated cancer cell death.

2.       In Figure 1C, what’s the cell line used?, and a rescue experiment is needed.

3.       In figure 2A, the upper two panels were supposed to be western blot data, what are the lower two panels? The data is not very clearly annotated.

4.       In figure 2B,C, the values in the graph represent three independent samples, so where are the error bars of each data point?

5.       In line 70, the authors claimed “surviving protein stability is modulated by the expression level of USP2”, but the data presented here doesn’t support this statement. Authors should also check the survivin protein level change with ectopic expression of USP2 wildtype and catalytic mutant.

6.       In Figure 5E, treatment of ML364 and TRAIL leads to lower protein level of endogenous surviving, but the ectopic protein expression level of surviving seems to be unaffected, could authors give a good explanation on this?

7.       In figure 6B, a control experiment in which ectopic expression of catalytic dead USP2 is needed.

the scientific accuracy of the manuscript is not enough, extensive editing of the English language required

Author Response

Dear, Editor
We sincerely appreciate the time and effort of you and the referees spent in considering and evaluating our manuscript (ijms-2488263) entitled with “Inhibition of USP2 Enhances TRAIL-Mediated Cancer Cell Death Through Downregulation of Survivin” by Tak Gyeom Lee et al., for publication in International Journal of Molecular Sciences. Having received the kind comments by reviewer, we revised our manuscript with attention to each of the comments by reviewer. We appreciate the reviewer very much, who raised the very important critiques to strengthen the claim of our manuscript. We have given very careful consideration to the suggestions and have revised our manuscript. We performed additional experiments and new information are incorporated in the revised version of our manuscript. We have responded all the comments by the referee point-by-point as follows.

Could find the attachment files?

Sincerely yours,

Reviewer 2 Report

The small signaling protein ubiquitin is ligated to cellular proteins by E3 ligase machinery that is both highly specific and tightly regulated. Specialized cellular machines called deubiquitinases are able to remove ubiquitin from proteins, often also with regulation and specificity. As ubiquitin frequently signals destruction of the protein, deubiquitinases are able to prevent degradation by the ubiquitin-proteasome system and therefore stabilize their target proteins. With over 300 E3 ligases and even more different targeting complexes, 100 deubiquitinases and 10000s of target proteins, working out which ligase and DUB control which part of which pathway is a monumental task.

The authors present evidence that deubiquitinase USP2 stabilizes survivin in several cancer cell lines. They show that USP2 and survivin physically interact and that downregulation of survivin does not originate in changes to mRNA expression or protein synthesis. They show evidence for ubiquitination of survivin in cell lines, which is more pronounced when USP2 is inhibited by specific small molecule inhibitor ML364. Using Ml364 and TRAIL in combination, they test apoptosis induction in cancer cell lines and come to the conclusion that USP2 has an essential function in TRAIL-mediated apoptosis. The authors use predominantly classic cell biology techniques, eschewing modern systems biology approaches like e.g. SILAC that are able to report changes to all proteins in the cell rather than a few selected proteins. Nevertheless, due to diligent study design and appropriately chose controls, the authors are able to identify and confirm the involvement of USP2 in apoptosis. These results are novel and valuable and fully deserve publication.

The overall rigorous and lucid presentation is unfortunately marred by severe language issues. I fully sympathize with the authors, having learned English as a second language myself, but these issues need to be addressed before the paper can be published.

Major issues:

Why was the cell line Caki-1 chosen for the experiments? It would be beneficial to briefly explain the reasoning behind the choice in the paper. The authors show that downregulation of survivin by ML264 is observed in other cancer cell lines as well, which is a great experiment. The degree to which it is downregulated is very different however. Do the authors have any indications for the mechanism behind this difference?

Minor/typographic issues:

229: would the conclusion be better presented before the Materials and Methods or is this the intended structure?

245: the entire text block appears to be unmodified template text. Should this be removed?

Author Response

(The authors gave the same response as above.)

Round 2

Reviewer 1 Report

my comments have been well-addressed, it can be accepted in present format